# Dementia in an Acute Hospital Setting: Health Service Research to Profile Patient Characteristics and Predictors of Adverse Clinical Outcomes

**DOI:** 10.3390/geriatrics4010007

**Published:** 2019-01-02

**Authors:** Inderpal Singh, Chris Edwards, Daniel Duric, Aman Rasuly, Sabdat Oziohu Musa, Anser Anwar

**Affiliations:** 1Consultant Geriatrician, Department of Geriatric Medicine, Ysbyty Ystrad Fawr, Aneurin Bevan University Health Board, Wales CF82 7EP, UK; 2Department of Dermatology, St Wollas Hospital, Aneurin Bevan University Health Board, Newport NP20 4SZ, UK; Chris.Edwards3@wales.nhs.uk; 3Speciality Doctor, Department of Geriatric Medicine, Ysbyty Ystrad Fawr, Aneurin Bevan University Health Board, Wales CF82 7EP, UK; Daniel.Duric@wales.nhs.uk (D.D.); Amanullah.Rasuly@wales.nhs.uk (A.R.); Sabdat.Musa@wales.nhs.uk (S.O.M.); Anser.Anwar@wales.nhs.uk (A.A.)

**Keywords:** dementia, clinical outcomes, mortality, falls, acute older people, hospital

## Abstract

Introduction: Patients with dementia often have other associated medical co-morbidities resulting in adverse outcomes. The National Audit of Dementia (NAD) in the UK showed a wide variation in the quality and clinical care for acute dementia patients. This study aims to record the clinical profile and benchmark clinical outcomes of acute dementia patients admitted within Aneurin Bevan University Health Board, Wales (UK). Methods: This was a retrospective observational study based on analysis of the existing data for all acute dementia patients. Ethical approval was not required for this service evaluation. Results: In 2016, a total of 1770 dementia patients had 2474 acute admissions. We studied 1167 acute admissions (953 dementia patients) from 1st January 2016–30th June 2016. The mean age was 84.5 ± 7.8 years (females = 63.5%). Mean Charlson comorbidity index and the number of drugs were 6.0 ± 1.5 and 5.1 ± 2.1. 15.4% (147/953) patients were on antipsychotics. Overall mean hospital stay was 19.4 ± 27.2 days. 30-days readmission rate was 17.2% (138/800) with a mean hospital stay of 14.6 ± 17.9 days. 3.4% (32/953) patients were excluded due to a coding error. 70.3% (n = 670/953) were previously living in their own homes and only 26.3% (n = 251/953) were admitted from care homes. 59.5% patients (n = 399/670) were discharged back to their homes and 21.6% (145/670) were discharged to a new care home, which represents an approximately 1.68 times higher rate of new care home occupancy than the patients being originally admitted from a care home. Overall inpatient was 16.0% (153/953). 30-days and one-year mortality were 22.3% (213/953) and 49.2% (469/953) respectively. The observed mortality rates between patients admitted from home or from a care home were highly significant for one-year mortality (*p* < 0.001). The inpatient falls rate was significantly higher (1.8 times) as compared to overall general medical inpatient falls rate. Conclusion: Acute patients with dementia have a higher risk of adverse outcomes and the impact of hospitalisation. Prompt comprehensive geriatric assessment and quality improvement initiatives are needed to improve clinical outcomes and to enhance the quality of care.

## 1. Introduction

NHS is facing an ageing population and older people have a higher prevalence of mental health problems in the general hospital settings as compared to the community. More than a third of patients older than 70 years requiring acute admission have dementia, but only half of these patients have been diagnosed [1,2].

Patients with dementia often have other associated medical co-morbidities which directly or indirectly could result in poorer outcomes including inpatient falls, longer hospital stays, emergency readmission and institutionalisation [2,3,4,5,6]. Admission to an acute hospital can be distressing and disorientating for a person with dementia and is associated with a decline in cognitive and functional ability.

The National Audit of Dementia (NAD) in the UK showed a wide variation in the quality and approach of care for acutely unwell patients with dementia admitted to the hospital [7]. Dementia care relating to multidisciplinary assessment, staffing levels, staff education and training; rehabilitation and discharge planning; and access to certain specialist services including Psychiatry Liaison service can be variable and sub-optimal [8]. This could potentially increase the risk of adverse clinical outcomes.

The objective of this study is to record the demographics, and patient characteristics including medical co-morbidities and polypharmacy, to understand and benchmark the clinical outcomes of those patients with dementia admitted acutely across three acute sites within Aneurin Bevan University Health Board (ABUHB). This would help us to explore quality initiatives to improve our patient care and plan our future services accordingly. The secondary objective is to study the predictors of poor outcome including 30-day readmission rate, inpatient mortality and discharge to a new care home.

## 2. Methods

### 2.1. Study Design

This was a retrospective observational cohort study based on analysis of the existing data for all the patients with dementia admitted acutely to ABUHB.

### 2.2. Setting

All patients with dementia admitted acutely to one of the three acute hospital sites (Royal Gwent Hospital, Nevill Hall Hospital and Ysbyty Ystrad Fawr) within ABUHB were included.

### 2.3. Data and Statistical Analysis

Information on demographics, medical co-morbidities, and clinical outcomes including inpatient fractures, discharge to a new care home, readmission and mortality was extracted electronically from the clinical workstation, clinic letters and coding from 1st January 2016 to 30th June 2016. Readmission and Mortality data was reviewed for 2 years until 30th June 2018. Data on inpatient falls was extracted from DATIX, which is a web-based patient safety software for healthcare risk management which includes incidents of inpatient falls. The description of the study cohort was completed and baseline characteristics of all patients have presented as means ± standard deviation. The *t*-tests were used to compare the mean baseline characteristics of patients, and chi-square tests were used to compare categorical variables. Mortality rates were compared using the difference between proportions test.

Sub-analysis for predictors of poor outcome (inpatient mortality, discharge to a new care home and readmission within 30 days) were explored using multivariate analysis for age, gender, Charlson Comorbidity Index (CCI), number of drugs, number of ward transfers, length of stay any fragility fracture (including hip, wrist, spine, humerus and pelvis) were studied using a Binomial Linear Model with logit link function.

We have considered 7 separate factors that we believe may be predictors of adverse outcomes. These can act independently or may interact with other factors in determining the outcomes, so the multivariate analysis is somewhat complex. The multivariate analysis results in likelihood scores for each factor or combination of factors.

All statistics were conducted using STATISTICA StatSoft data analysis software system, version 9.1 (StatSoft, Inc., 2010, Tulsa, OK, USA). The generalised linear and non-linear models building module of the Statistica statistics package used binomial modelling with logit linking. *p*-values ≤ 0.05 were taken to be statistically significant.

Ethical approval was not required for this service evaluation as this work does not constitute a research study according to the Health Research Authority decision tool. In addition, this service evaluation was completed based on the recommendations of the NAD, which is a clinical audit programme, looking at the quality of care received by people with dementia in general hospitals. However, all questions and forms required to carry out the study were sent to the ABUHB Research and Development (R&D) Department, to assess risks to patient identification and the Health Board. The study was approved by the R&D as a service evaluation as patients were not directly interviewed and no identifiable patient data was recorded. The study was compliant with the personal data protection regulation in the U.K.

## 3. Results

### 3.1. Patient Characteristics

The total number of emergency admissions above 18 years and above 65 years across three acute sites in the whole year 2016 were 37378 and 21437 respectively. A total of 2474 acute admission were recorded in the year 2016 from the 1770 acute dementia patients. We studied 953 consecutive dementia patients from 01st January 2016 to 30th June 2016 who had a total of 1167 episodes of acute admissions.

The mean age on admission was 84.5 ± 7.8 years. The proportion of females was 63.5% and the mean age of females (85.2 ± 7.6 years) was significantly higher in comparison to the mean age of males (82.0 ± 7.7 years) (*p* < 0.001).

About 70% (n = 670/953) were previously living in their own homes and 26.34% (n = 251/953) were admitted from care homes. Mean CCI and the number of drugs was 6.05 ± 1.5 and 5.1 ± 2.1 respectively. 15.4% (147/953) patients were on antipsychotics. 6% of patients had 3 more or transfer during an index admission.

### 3.2. Clinical Outcomes

Overall mean hospital stay was 19.4 ± 27.2 days (median 9 days; range 0–326 days). 3.4% (32/953) patients were excluded from analysis for discharge destination due to coding errors noted in the Clinical Work Station. 59.5% patients admitted from their own homes (n = 399/670) were discharged back to the original residence, 21.6% (145/670) were discharged to a new care home and 18.1% (121/670) died. Among those admitted from care homes, 79.7% (200/251) patients were discharged back to original care home and 20.3% patients (51/251) died. The rate of discharge to a new care home was an approximately 1.68 times higher rate than the patients being originally admitted from a care home (251 referred from a care home but 51 died, 145 discharged to a new care home, therefore total CH discharges = 345).

Overall 1.6% patients (15/953) died within one day, 2.2% (21/953) died within 2 days, 2.4% patients died (23/953) within 3 days and 51 (5.3%) died within 7 days. However, 4.4% patients (11/251) admitted from care home died within 3 days as compared to 1.7% (12/690) admitted from the community, which is significantly higher (*p* = 0.017, a difference in proportion test). Overall inpatient mortality was 16.0% (153/953). The 30-days, 90-days and one-year mortality were 22.3% (213/953), 29.6% (283/953) and 49.2% (469/953) respectively (Figure 1). The mortality rate of those admitted form care homes as inpatient and at one year was 20.3% (51/251) and 66.1% (166/251) respectively. In comparison, the mortality rate of those admitted from own home as inpatient and at one year was 18.1% (12/670) and 43.6% (292/670) respectively. The observed mortality rates between patients admitted from home or from a care home were not significant for inpatient mortality (*p* = 0.45) but were highly significant for one-year mortality (*p* < 0.001), both using a difference in proportions test. Kaplan–Meier survival curve is shown in Figure 2 to show the proportion of patients with acute dementia living over two years or more.

Overall 30-day readmission rate was 17.2% (138/800) with a mean hospital stay of 14.6 ± 17.9 days. Nearly half of dementia patients (49.4%, 395/800) were readmitted once over one year. Patients admitted two and three times over two years follow up were 32% (259/800) and 17% (138/800) respectively.

The total number of inpatient falls in ABUHB across three acute sites including rehabilitation community hospital beds between 1st January 2016 and 30th June 2016 were 1799. The total occupied bed days (OBD) were 225812, giving the rate of inpatient falls of 7.96/1000 OBD. The rate of falls in general medical beds under unscheduled care during this period were 9.0/1000 OBD and in the mental health division were 9.9/1000 OBD.

The total number of falls for acute dementia patients during the same period of 6 months was 267, affecting 109 patients. The range of falls were 1 to 17 and mean inpatient falls among those with dementia was 2.44. The inpatient falls rate for dementia patients, based on the 18,515 bed days was 14.4/1000 OBD which is significantly higher (1.8 times) as compared to overall general medical beds inpatient falls rate (*p* < 0.0001, the difference in proportions test).

5% dementia patients (48/953) sustained a fracture following an inpatient fall during an index admission. Half (24/48) of inpatient fractures were the hip fracture. The incidence of inpatient hip fracture was 2.5% (n = 24/953). Other fractures were pelvis = 5; wrist = 2; humerus = 2; vertebrae = 1; ankle = 2; others = 12.

### 3.3. Predictors of Poor Outcomes

Sub-analysis were completed to appraise predictors of adverse outcomes. For the risk of inpatient mortality, the best prediction required all factors plus the interaction between age and CCI, with a likelihood score of 23.6. The most predictive single factor was age, with a likelihood score of 9.6. The best two predicting factors were age and the length of stay with a likelihood score of 15.26, and the best three factors were age, length of stay and presence of fragility fracture with a likelihood score of 19.0. These three factors accounted for 80% of the maximum likelihood of the risk.

For the risk of discharge to a new care home, the best prediction required all factors plus the interaction between age and CCI, with a likelihood score of 68.98. The most predictive single factor was the length of stay, with a likelihood score of 55.9. The best two predicting factors were age and the length of stay with a likelihood score of 67.8, and the best three factors were age, length of stay and number of ward transfers with a likelihood score of 68.5. These three factors accounted for 99% of the maximum likelihood of the risk.

For the risk of readmission within 30 days, the best prediction required all factors plus the interaction between age and CCI, with a likelihood score of 15.8. The most predictive single factor was the number of drugs, with a likelihood score of 11.5. The best two predicting factors were the number of drugs and the length of stay with a likelihood score of 13.5, and the best three factors were the number of drugs, length of stay and CCI with a likelihood score of 14.8. These three factors accounted for almost 94% of the maximum likelihood of the risk. 

## 4. Discussion

Worldwide, the numbers of people living with dementia will increase from 50 million in 2018 to 152 million in 2050, a 204% increase [9]. More than 850,000 people are affected in the UK and the number of patients with dementia will increase rapidly with the ageing population [10]. This rising prevalence of dementia will have a significant impact on health and social care costs.

Over 70% of inpatients are over the age of 65 years and caring for acutely unwell older patients is challenging for healthcare systems [1]. 30% of all inpatients in general hospitals could have dementia, who are particularly more challenging due to multiple co-morbidities, inability to communicate their care needs and atypical presentation [1]. People with dementia have the right to fair and equitable treatment but care needs may not be easily recognised and are often misinterpreted to be manifestations of dementia itself [11]. Therefore, caring for people with dementia inevitably places pressure on hospitals to provide safe high-quality care.

The risk of inpatient death in patients with dementia is higher (11.8%) as compared to those without dementia (6.6%) [5]. Delayed or under-recognition of dementia among patients by the medical staff is one the key reason for poorer clinical outcomes. In this study, inpatient mortality was 16%. A similar prospective observational study carried out in an urban tertiary Irish referral centre has also reported inpatient mortality of 15% [2].

We have observed that one-year mortality was higher for care home patients (66.1%) as compared to those form their own homes (43.6%). One-year mortality in older care home residents in England and Wales has been reported as 26.2% but we are not aware of any study which has reported mortality for hospitalised acute dementia patients from care home [12]. In this study, overall 2% patients died within the first two days of the hospital admission. The proportion of patients dying within the first 3 days was significantly higher for those admitted from a care home. This does not only warrants further studies but suggest an enhanced integrated partnership working among old age psychiatry, care of the elderly and community teams to consider advanced care planning and supporting last days of life in the preferred place of residence.

Dementia is associated with impaired mobility and people with dementia are at two times higher risk of inpatient fall and adverse outcomes including discharge to a new care home and prolonged length of stay when compared to those with no cognitive impairment [3,13]. In this study, inpatient falls rate for acute dementia patients was 1.8 times as compared to all patients admitted during 6 months.

Patients with dementia are at a 3-fold increase risk of preventable readmissions and require more healthcare services than those without dementia [14]. The 30-day readmission rate in this study was 17.2% but half of the patients were readmitted over one-year.

The point prevalence of acute delirium among inpatients has been reported between 17.6% and 20.7% using different diagnostic criteria in the same population, with a prevalence of 34.8% above 80 years [15]. A meta-analysis based on pooled findings from multiple small studies has reported a prevalence of delirium on admission in 10–31% of medical inpatients and 3–29% during hospital stay could develop acute delirium [16]. There is a dearth of definite and accurate data from the hospitals or Health Board over a longer period. In this study, documented delirium was mentioned in the 9.6% of e-discharges. Based on NAD report 2016–2017, it is likely that delirium in this study has been under recognised and under-reported. Nevertheless, the prescription of antipsychotic drugs in 15.6% of dementia patients might suggest that these patients were receiving treatment for delirium. Therefore, a definitive and accurate determination of in-hospital delirium prevalence, using standardized delirium instruments, is needed. Previous research found that smartphone application can enhance doctors’ awareness of delirium and enhance diagnostic accuracy [17]. This will augment treating delirium based on the current evidence and recommendations [18].

The best predicting factors for poor outcomes including inpatient mortality, discharge to a new care home or readmission within 30 days were age, length of stay, co-morbidity burden and number of drugs. The most predictive single factor for inpatient mortality was age as shown previously [19].

This study has several strengths. It is in line with the recommendations of NAD and would support health care organisations to establish a pragmatic approach to record such clinical outcomes as part of clinical governance. We have explored wider clinical outcomes for acutely unwell dementia patients including readmission and long-term mortality. Impact of hospitalisation was explored with respect to inpatient falls. This study has also analysed predictors of poor outcomes which could help clinical staff to streamline care plan towards the end of life priorities and supporting families and carers to formulate advance care plans, maintaining dignity during last days of life. We learnt that the Health Board does not keep separate complaint data for dementia patients and it was not pragmatic to analyse complaints data. An effort has been taken to benchmark the existing clinical outcome to streamline the processes for reporting dynamic data to Dementia Board and Executive team on regular basis. This could lead to an introduction of quality initiatives to not only measure but minimise the impact of hospitalisation, particularly inpatient falls, dehydration and pressure ulcers.

We acknowledge the low power of this study as a weakness of the study. We also acknowledge that the data is only from one Health Board, therefore it cannot be generalised and we have not examined the comprehensive impact of hospitalisation including delirium, dehydration and pressure sores. This study has not completed an in-depth analysis for reasons for readmission or cause of death. We also acknowledge that this is a retrospective study, therefore data on type of dementia, severity of dementia, caregiver burden, carers’ views, health-related quality of life, work-related staff stress and the number of patients requiring one-to-one care is not being reported. Furthermore, this study did not report the prevalence of stroke [20] and depression [21] which are commonly associated with dementia and lead to adverse outcomes. Further studies to overcome these limitations is warranted.

With the growing number of people with dementia and associated adverse clinical outcomes, training of our future healthcare professionals in the care of older people is needed. Lack of appropriate training to complete dementia assessment and discharge planning could be a factor for sub-optimal care and poor clinical outcomes [8]. Nursing staff training on geriatric giants have shown to reduce work-related stress and similarly, dementia teaching to medical students and foundation doctors have shown to improve competencies but the impact of such interventions on clinical outcomes has not been studied yet [22,23,24]. Psychiatry Liaison services like Rapid Assessment Interface and Discharge (RAID) have shown improved clinical outcomes for acutely unwell patients with mental health problems [25,26,27]. Further investment and developing more links with the community services could possibly avoid admitting very frail patients to the hospital, particularly those with advanced dementia.

The standard recommendations are to commission dynamic research using a fixed methodology to estimate changes in dementia incidence, prevalence and particularly mortality over time [28]. However, there is a dearth of research that tracks all three parameters. Regular evaluation of clinical outcome data to benchmark services for acute older people has been proposed by the NAD [7]. Such data could be used to introduce quality improvement initiatives to improve patient-related clinical outcomes in this vulnerable group [29].

## 5. Conclusions

Acute patients with dementia have a higher risk of adverse outcomes and the impact of hospitalisation. Improving safety and quality of care for patients with dementia in acute hospitals will benefit all patients and is a high priority for the NHS. Further similar studies will improve organizational understanding of clinical outcomes for acute dementia patients. This would also facilitate quality improvement initiatives to improve patient care and modernisation of community service.

## Figures and Tables

**Figure 1 geriatrics-04-00007-f001:**
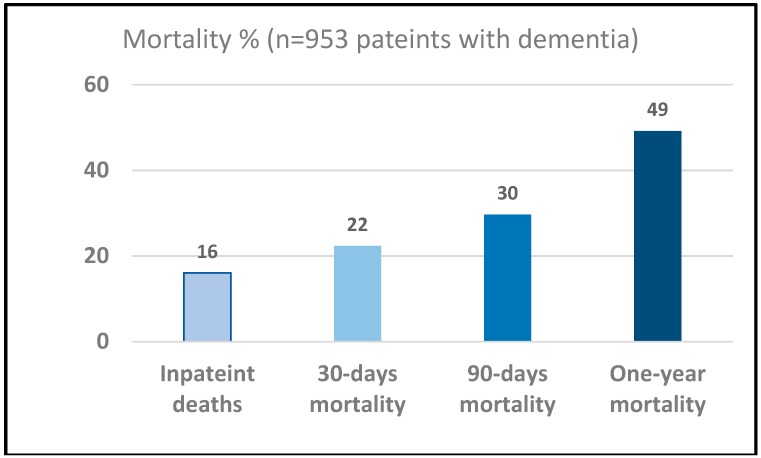
Mortality outcome (%)—Inpatient, 30 days, 90 days and one-year.

**Figure 2 geriatrics-04-00007-f002:**
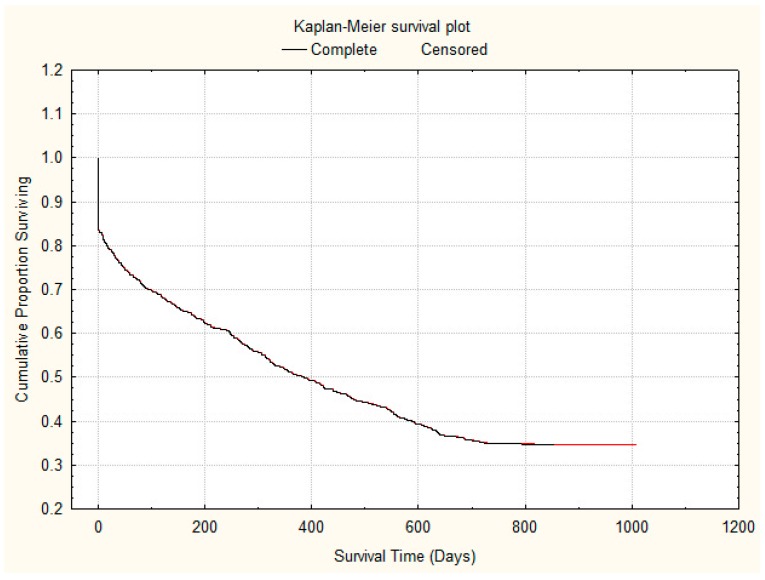
Kaplein-Meier showing survival over days during follow-up period.

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
