# Peer review of "Dementia in an Acute Hospital Setting: Health Service Research to Profile Patient Characteristics and Predictors of Adverse Clinical Outcomes"

_geriatrics, 2019, doi:10.3390/geriatrics4010007_

Reviewer 1 Report

Dear Authors

The topic of the paper is relevant and well described.

Only two comments:

Could cause of dead be reported?

Is there information of stage of dementia?

Dear Authors

The topic of the paper is relevant and well described.

Only two comments:

Could cause of dead be reported?

Response: This was not recorded and has been mentioned as a limittaion of the study.

Is there information of stage of dementia?

Response: This was not recorded as this was retrospective study and has been mentioned as a limittaion of the study.

I Singh

Reviewer 2 Report

Thank you for inviting me to review the paper on “Dementia in an acute hospital setting: health service research to profile patient characteristics and predictors of adverse clinical outcomes”. This is an important paper and I recommend publication in Geriatrics. I have the following recommendations:

1.     Although the authors stated that this study did not reply ethics approval, I recommend the authors to add a statement to indicate that this study is compliant with personal data protection regulation in the U.K. Please add the following statement on Pg 3 line 93.

… data was recorded. The study was compliant with the personal data protection regulation in the U.K.

2.     Line 104, the authors stated that “70.3% (n = 670/953) were previously living in their own homes.” A sentence should not start with number. Please change to About 70% (n=670/953) participants ….

3.     Line 212, the authors stated that “it is likely that delirium is this study has been under recognised and under-reported.” It should be delirium “in” this study.

4.     Please link the usage of antipsychotic drugs and delirium. Please add the following statement:

Line 212 … under-reported. Nevertheless, the prescription of antipsychotic drugs in 15.6% of dementia patients might suggest that these patients were receiving treatment for delirium.

5.     I suggest adding the new initiatives to enhance doctors’ awareness of delirium. Please add the following statement.

Line 214 …. instruments, is needed. Previous research found that smartphone application can enhance doctors’ awareness of delirium and enhance diagnostic accuracy.  (Reference: PMID: 25735309).

6.     Under limitations, please add that this study did not report the prevalence of depression and stroke which are important adverse comorbidity. Please add the following statement:

Line 238 … is not being reported. Furthermore, this study did not report the prevalence of stroke (Reference PMID: 27533593) and depression (Reference PMID: 23236014) which are commonly associated with dementia and lead to adverse outcomes.

Author Response

Thank you for inviting me to review the paper on “Dementia in an acute hospital setting: health service research to profile patient characteristics and predictors of adverse clinical outcomes”. This is an important paper and I recommend publication in Geriatrics. I have the following recommendations:

1.     Although the authors stated that this study did not reply ethics approval, I recommend the authors to add a statement to indicate that this study is compliant with personal data protection regulation in the U.K. Please add the following statement on Pg 3 line 93.

data was recorded. The study was compliant with the personal data protection regulation in the U.K.

Response: We have now mentioned this in the manuscript.

2.     Line 104, the authors stated that “70.3% (n = 670/953) were previously living in their own homes.” A sentence should not start with number. Please change to About 70% (n=670/953) participants ….

Response: Correction has been made.

3.     Line 212, the authors stated that “it is likely that delirium is this study has been under recognised and under-reported.” It should be delirium “in” this study.

Response: Thanks for bringing this typo error to our attention.

4.     Please link the usage of antipsychotic drugs and delirium. Please add the following statement:

Line 212 … under-reported. Nevertheless, the prescription of antipsychotic drugs in 15.6% of dementia patients might suggest that these patients were receiving treatment for delirium.

Response: Thanks for your suggetsion, we have added this statement.

5.     I suggest adding the new initiatives to enhance doctors’ awareness of delirium. Please add the following statement.

Line 214 …. instruments, is needed. Previous research found that smartphone application can enhance doctors’ awareness of delirium and enhance diagnostic accuracy.  (Reference: PMID: 25735309).

Response: We agree with your suggestion and this new reference has been added.

6.     Under limitations, please add that this study did not report the prevalence of depression and stroke which are important adverse comorbidity. Please add the following statement:

Line 238 … is not being reported. Furthermore, this study did not report the prevalence of stroke (Reference PMID: 27533593) and depression (Reference PMID: 23236014) which are commonly associated with dementia and lead to adverse outcomes.

Response: We have mentioned this unedr limitation section that this study did not report prevalence of stroke and depression with references.

All authors thank and acknowledge reviewers comments and have been very grateful to them for taking time and giving us very constructive and useful comments.

Thanks

Dr I Singh